# Inferring feature importance with uncertainties with application to large genotype data

**Pål Vegard Johnsen**[1,2]*, **Inga Strümke**[3,4], **Mette Langaas**[2], **Andrew Thomas DeWan**[5], **Signe Riemer-Sørensen**[1]

**1** SINTEF DIGITAL, Oslo, Norway, **2** Department of Mathematical Sciences, Norwegian University of Science and Technology, Trondheim, Norway, **3** Department of Engineering Cybernetics, Norwegian University of Science and Technology, Trondheim, Norway, **4** Department of Holistic Systems, SimulaMet, Oslo, Norway, **5** Department of Chronic Disease Epidemiology and Center for Perinatal, Pediatric and Environmental Epidemiology, Yale School of Public Health, New Haven, Connecticut, United States of America

* pal.johnsen@sintef.no

**Data Availability Statement:** UK Biobank data can be obtained from UK Biobank project site, subject to successful registration and application process. Further details can be found at https://www.

## Abstract

Estimating feature importance, which is the contribution of a prediction or several predictions due to a feature, is an essential aspect of explaining data-based models. Besides explaining the model itself, an equally relevant question is which features are important in the underlying data generating process. We present a Shapley-value-based framework for inferring the importance of individual features, including uncertainty in the estimator. We build upon the recently published model-agnostic feature importance score of SAGE (Shapley additive global importance) and introduce Sub-SAGE. For tree-based models, it has the advantage that it can be estimated without computationally expensive resampling. We argue that for all model types the uncertainties in our Sub-SAGE estimator can be estimated using bootstrapping and demonstrate the approach for tree ensemble methods. The framework is exemplified on synthetic data as well as large genotype data for predicting feature importance with respect to obesity.

## Author summary

Artificial intelligence and machine learning have been increasingly popular tools for modelling complex relationships in medicine and genomics. For example a machine learning model for predicting the likelihood of a particular person developing some disease. The prediction model can for instance be based on genomics data, which consists of a large number of features for each single person. Such prediction models can be very complex and difficult to interpret, hence they are often denoted black-box models. However, to exploit the knowledge the prediction model has gained, we must be able to interpret it, and explain which features are important for the model, but also for the underlying data. We investigate a theoretical approach for extracting feature importance, even when the model input consists of many features. Lastly, we emphasize the need for

ukbiobank.ac.uk/enable-your-research/apply-for-access Source code, including numerical values and scripts for providing the figures in this paper can be found online at: https://github.com/palVJ/subSAGE.

**Funding:** This research was funded by The Research Council of Norway (https://www.forskningsradet.no/en/), Grant 272402, Ph.D. Scholarship at SINTEF, including funding for a research stay abroad at Yale School of Public Health to PVJ. The funders had no role in study design, data collection and analysis, decision to publish, or preparation of the manuscript.

**Competing interests:** The authors declare that they have no competing interests.

estimating the uncertainty of the individual feature importance, and provide a bootstrap procedure for doing so.

## Introduction

With the strong improvement of black-box machine learning models such as gradient boosting models and deep neural networks, the question of how to infer feature importance, including uncertainty estimates, in these types of models has become increasingly important. This is particularly important if the results from the model can be trustfully investigated further within medicine or genomics such as when applied to drug discovery [1]. The Shapley decomposition, a solution concept from cooperative game theory [2], has enjoyed a surge of interest in the literature on explainable artificial intelligence in recent years, (cf. [3–16]). A widely used Shapley-based framework for deriving feature importance in machine learning models post-training is Shapley additive explanations (SHAP) [4, 6], which explains individual predictions' deviations from the average model prediction. As such, SHAP attributes feature importance as they are perceived by the *model*. The more recently introduced Shapley additive global importance (SAGE) is also based on the Shapley decomposition, but attributes feature importance by a global decomposition of the model loss across a whole data set [17]. The SAGE framework thus provides an explanation of the influence of the features taking into account not only the model, but also implicitly the data via the loss function, thus encapsulating that the model might not be—and most likely is not—a perfect description of the data [18].

The SAGE value needs to be estimated, and the SAGE estimator is itself a random variable as the corresponding SAGE estimate is based on data of finite size generated from some unknown probability distribution. As is the case for any feature importance score, we argue that the uncertainty in the estimate is equally important as the estimate itself for drawing conclusions. However, computation of the SAGE-estimate is infeasible even for moderate-sized data, and thus further approximations are needed [17]. To this end, we introduce Sub-SAGE, which is motivated by SAGE but can be estimated exactly for tree-ensemble models, by using a reduced subset of coalitions. Additionally, we describe how to estimate a confidence interval for the Sub-SAGE value. No calculation of such uncertainty exists in the SAGE package or the literature. We estimate the confidence interval using paired bootstrapping, and demonstrate its calculation for tree ensemble models on simulated as well as genotype data. We argue that this procedure provides a way to infer the true feature importance in the underlying data. The remainder of this paper is structured as follows. In Materials and methods we introduce the particular genotype data set to be used, as well as background concepts such as the Shapley value, SHAP and SAGE, before moving on to Sub-SAGE, and its uncertainty. The method is exemplified using synthetic data. In the Results section, the method is applied on the genotype data before we discuss the results in Discussion and conclusion.

## Materials and methods

### Data and use case

In order to evaluate the Sub-SAGE feature importance score, we will apply it using collected genotype data from the UK Biobank [19, 20]. UK Biobank is a large prospective cohort study in the United Kingdom that began in 2006 consisting of about 500000 participants. As use

case, we considered the aim of inferring the feature importance of single nucleotide polymorphisms (SNPs) with respect to a logistic regression model for predicting the susceptibility of obesity (BMI $\geq$ 30). The model includes a large number of SNPs as features, as well as accounting for non-linear effects. Obesity was selected since this particular trait has been extensively researched in previous genome-wide association studies (GWAS) providing a meaningful way to evaluate our method [21].

## Ethics statement

Ethical approval was obtained by the UK Biobank from the North West Multicentre Research Ethics Committee, the National Information Governance Board for Health and Social Care in England and Wales, and the Community Health Index Advisory Group in Scotland. All participants provided written informed consent. The research in this paper has been conducted using the UK Biobank Resource under Application Number 32285. The application for access to the UK Biobank Resource was approved on October 10, 2018.

## Shapley-based explanation methods

We provide a brief introduction to the Shapley decomposition-based SHAP and SAGE frameworks to quantify feature importance in machine learning models. An advantage of such Shapley-based frameworks is that they in principle can be applied on any input-output model, parametric or non-parametric, including non-linear machine learning models such as neural networks or tree ensemble models. Consequently, non-linear effects can also be captured using these frameworks. The Shapley decomposition is a solution concept from cooperative game theory [2]. It provides a decomposition of *any* value function $v(\mathcal{S})$ that characterises the game, and produces a single real number, or payoff, per set of players in the game (coalitions). The resulting decomposition satisfies the three properties of efficiency, monotonicity and symmetry, and is provably the only method to satisfy all three [22, 23, Thm. 2]. For details see S1 File.

Consider a supervised learning task characterised by a set of $M$ features $\mathbf{x}_i$ and corresponding univariate responses $y_i$, for $i = 1, \ldots, N$, and a fitted model that is a mapping from feature values to response values, i.e. $\mathbf{x}_i \rightarrow \hat{y}(\mathbf{x}_i)$. As usual, uppercase letters denote random variables while lowercase letters denote observed data values. In this work, we assume independent features, implying $E[X_j | X_k = x_k] = E[X_j] \, \forall j \neq k$. This assumption is further discussed in Discussion and conclusion section.

**The SHAP value. Definition 1**. Let $\mathcal{S} \subseteq \mathcal{M} \setminus \{k\}$, with $\mathcal{M} = \{1, \ldots, M\}$, denote a subset of all features not including feature $k$. $\overline{\mathcal{S}}$ denote the corresponding complement subset of excluded features ($\mathcal{S} \cup \overline{\mathcal{S}} = \mathcal{M}$). The SHAP value, $\phi_k^{\mathrm{SHAP}}(\mathbf{x}, \hat{y})$, for a feature with index $k$ with respect to feature values $\mathbf{x}$ and a corresponding fitted model $\hat{y}$, is defined as [6]

$$\phi_k^{\mathrm{SHAP}}(\mathbf{x}, \hat{y}) = \sum_{\mathcal{S} \subseteq \mathcal{M} \setminus \{k\}} \frac{|\mathcal{S}|!(M - |\mathcal{S}| - 1)!}{M!} \left[ v_{\mathbf{x}, \hat{y}}(\mathcal{S} \cup \{k\}) - v_{\mathbf{x}, \hat{y}}(\mathcal{S}) \right]. \tag{1}$$

Here, the value function $v_{\mathbf{x}, \hat{y}}(\mathcal{S})$ is defined as the expected output of a prediction model conditioned that only a subset $\mathcal{S}$ of all features are included in the model,

$$v_{\mathbf{x}, \hat{y}}(\mathcal{S}) = E_{\mathbf{x}_{\overline{\mathcal{S}}}}[\hat{y}(\mathbf{X} | \mathbf{X}_{\mathcal{S}} = \mathbf{x}_{\mathcal{S}})]. \tag{2}$$

For instance, if $\mathbf{X}_{\overline{\mathcal{S}}}$ is a continuous random vector and we assume all features to be mutually

independent, we have

$$
\begin{aligned}
E_{\mathbf{X}_{\overline{S}}}[\hat{y}(\mathbf{X}|\mathbf{X}_{S} = \mathbf{x}_{S})] \quad &= \int_{\mathbf{x}_{\overline{S}}} \hat{y}(\mathbf{X}_{S} = \mathbf{x}_{S}, \mathbf{X}_{\overline{S}} = \mathbf{x}_{\overline{S}}) p(\mathbf{X}_{\overline{S}} = \mathbf{x}_{\overline{S}}|\mathbf{X}_{S} = \mathbf{x}_{S}) d\mathbf{x}_{\overline{S}} \\
&= \int_{\mathbf{x}_{\overline{S}}} \hat{y}(\mathbf{X}_{S} = \mathbf{x}_{S}, \mathbf{X}_{\overline{S}} = \mathbf{x}_{\overline{S}}) p(\mathbf{X}_{\overline{S}} = \mathbf{x}_{\overline{S}}) d\mathbf{x}_{\overline{S}} \, .
\end{aligned}
\tag{3}
$$

The stochastic behaviour in $\hat{y}(\mathbf{X}|\mathbf{X}_{S} = \mathbf{x}_{S})$ is due to the random vector $\mathbf{X}_{\overline{S}}$ of unknown feature values. We can think of the difference $v_{\mathbf{x},\hat{y}}(S \cup \{k\}) - v_{\mathbf{x},\hat{y}}(S)$ as the mean difference in a single model prediction when using feature $k$ in the model compared to when the value of feature $k$ is absent. The larger absolute SHAP value a feature $k$ has in a single prediction, the more influence the feature is regarded to have on this particular prediction.

**The SAGE value.** Define a loss function $\ell(y_i, \hat{y}(\mathbf{x}_i))$ as a measure of how well the fitted model $\hat{y}(\mathbf{x}_i)$ maps the features to a response, compared to the true response value $y_i$. As defined in [17], we take the SAGE value function $w(S)$ as the expected difference in the observed value of the loss function when the features in $S$ are included in the model compared to excluding all features.

**Definition 2**. Given a data generating process $(\mathbf{X}, Y)$, a function $\hat{y}$ to model the relationship between $\mathbf{X}$ and $Y$, and a loss function $\ell(y_i, \hat{y}(\mathbf{x}_i))$, we define $w_{\mathbf{X},Y,\hat{y}}(S)$ as:

$$
w_{\mathbf{X},Y,\hat{y}}(S) = E_{\mathbf{X},Y}[\ell(Y, V_{\mathbf{X},\hat{y}}(\emptyset))] - E_{\mathbf{X},Y}[\ell(Y, V_{\mathbf{X},\hat{y}}(S))] \, .
\tag{4}
$$

Here, $\emptyset$ denotes the empty set, while $V_{\mathbf{X},\hat{y}}(S)$ is the stochastic version of Eq (2). Specifically, $V_{\mathbf{X},\hat{y}}(S)$ is a random variable since its observed value varies depending on the random vector $X_S$. $v_{\mathbf{x},\hat{y}}(S)$ is a constant as we condition on the *observed* vector $\mathbf{x}_S$. For the case where $\mathbf{x}$ and $y$ are continuous, the expected value of the loss function when only a subset $S$ of feature values are known is given by

$$
E_{\mathbf{X},Y}[\ell(Y, V_{\mathbf{X},\hat{y}}(S))] = \int_y \int_{\mathbf{x}_S} \ell(y(\mathbf{x}), E_{\mathbf{X}_{\overline{S}}}[\hat{y}(\mathbf{X}|\mathbf{X}_S = \mathbf{x}_S)]) p(y|\mathbf{x}_S) p(\mathbf{x}_S) d\mathbf{x}_S dy \, .
\tag{5}
$$

Notice that the computation of $v_{\mathbf{x},\hat{y}}(S) = E_{\mathbf{X}_{\overline{S}}}[\hat{y}(\mathbf{X}|\mathbf{X}_S = \mathbf{x}_S)]$ happens inside the loss function, which is usually non-linear. Also note that in Eq (5), we integrate over *all* possible values of $X_S$.

**Definition 3**. The SAGE value for a feature $k$ is defined as [17]

$$
\phi_k^{\text{SAGE}}(\mathbf{X}, Y, \hat{y}) = \sum_{S \subseteq \mathcal{M} \setminus \{k\}} \frac{|S|!(M - |S| - 1)!}{M!} [w_{\mathbf{X},Y,\hat{y}}(S \cup \{k\}) - w_{\mathbf{X},Y,\hat{y}}(S)] \, .
\tag{6}
$$

We can think of the difference $w_{\mathbf{X},Y,\hat{y}}(S \cup \{k\}) - w_{\mathbf{X},Y,\hat{y}}(S)$ as the expected difference in the loss function when including feature $k$ in the model compared to excluding feature $k$ with respect to the subset $S$ of known feature values. SAGE is therefore a global feature importance score (as opposed to the local SHAP value) as it does not evaluate a single prediction, but rather the impact feature $k$ has across all predictions. The use of the loss function in the SAGE definition also ensures that the feature importance is not only based on the model, as for the SHAP value, but also on the data itself.

An interpretation of SAGE is that a positive SAGE value for a feature implies that including this feature in the model reduces the expected model loss compared to when not including the feature.

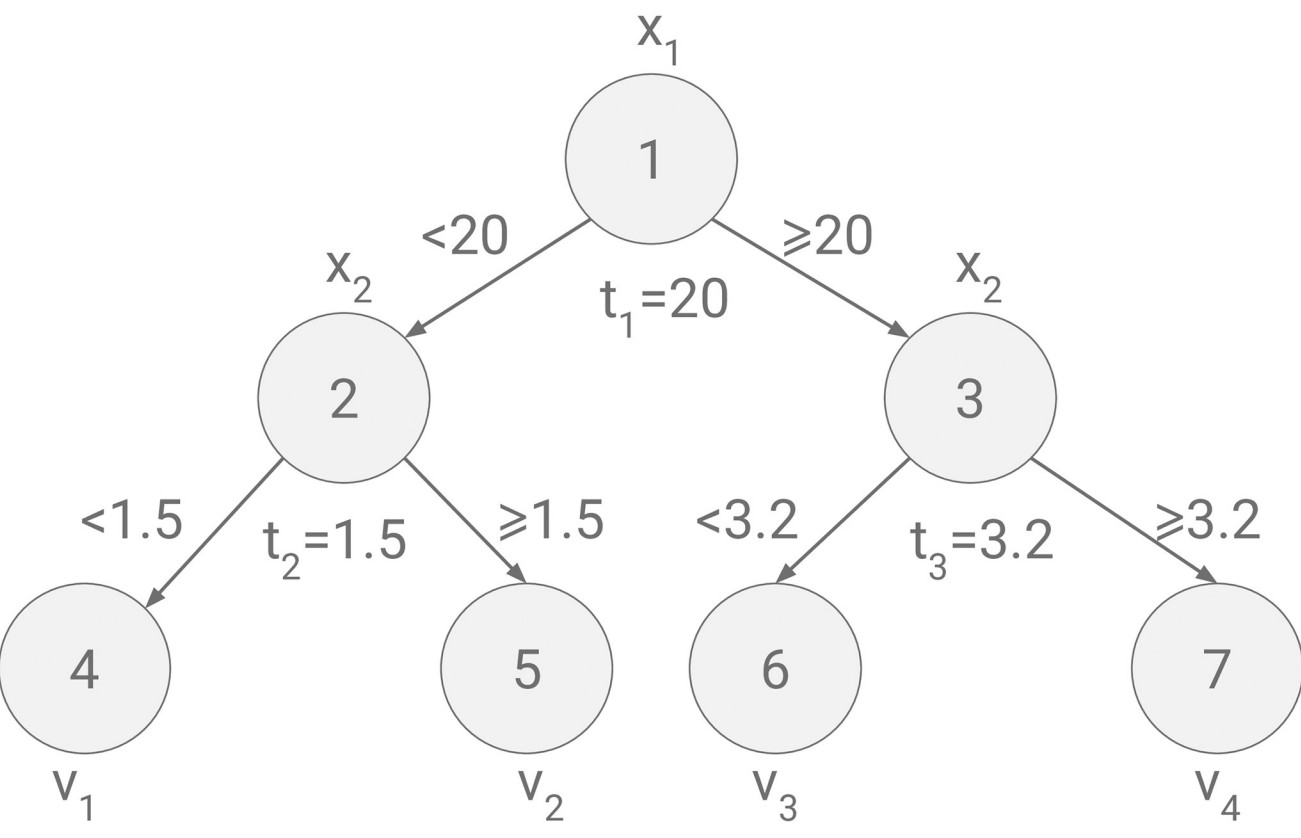

**Fig 1. A regression tree including two features $X_1$ and $X_2$.**

The features and response can be continuous or discrete. In the discrete case, integrals must be replaced by sums in Eqs (3) and (5). The expressions in Eqs. (2) and (4) are in general unknown and need to be estimated for each choice of model and loss function. Consequently, the SHAP and SAGE values become estimates as well.

**Tree ensemble models.** Consider a tree ensemble model consisting of several regression trees $f_\tau(\mathbf{x}_i)$ with predicted response $\hat{y}(\mathbf{x}_i)$, such that $\hat{y}(\mathbf{x}_i) = \sum_{\tau=1}^{T} f_\tau(\mathbf{x}_i)$ for $T$ trees. By the linearity property of the expected value, we have

$$v_{\mathbf{x},\hat{y}}(\mathcal{S}) = E_{\mathbf{X}_{\bar{\mathcal{S}}}}\left[\sum_{\tau=1}^{T} f_\tau(\mathbf{X}|\mathbf{X}_{\mathcal{S}} = \mathbf{x}_{\mathcal{S}})\right] = \sum_{\tau=1}^{T} E_{\mathbf{X}_{\bar{\mathcal{S}}}}[f_\tau(\mathbf{X}|\mathbf{X}_{\mathcal{S}} = \mathbf{x}_{\mathcal{S}})]. \quad (7)$$

The computation of $E_{\mathbf{X}_{\bar{\mathcal{S}}}}[f_\tau(\mathbf{X}|\mathbf{X}_{\mathcal{S}} = \mathbf{x}_{\mathcal{S}})]$ can be understood through a simple example: The regression tree illustrated in Fig 1 has depth two and splits on the two features indexed 1 and 2, which are continuous and mutually independent. The regression tree has parameters such as *splitting points*, $t_j$, for branch nodes, and *leaf values* $v_j$, for leaf nodes. For an observed value of $x_2 = 3$ we have

$$\begin{aligned} E_{\mathbf{X}_{\bar{\mathcal{S}}}}[f_\tau(\mathbf{X}|\mathbf{X}_{\mathcal{S}} = \mathbf{x}_{\mathcal{S}})] &= E_{X_1}[f_\tau(X_1|X_2 = 3)] \\ &= P(X_1 \geq 20)v_3 + P(X_1 < 20)v_2. \end{aligned} \quad (8)$$

In general, we do not know the value of $P(X_1 \leq 20)$, and need to estimate it. Consider $N$ data instances with recorded feature values from feature $k$. An *unbiased* estimate of $P(X_k \leq t)$ is then

$$\hat{P}(X_k \leq t) = \frac{1}{N}\sum_{i=1}^{N}I(x_{i,k} \leq t), \qquad (9)$$

where $x_{i,k}$ is the observed value of feature $k$ for data instance $i$, and $I(\cdot)$ is the indicator function. Using this estimate, we can also get an unbiased estimate for Eq (8). An unbiased estimate of $E_{\mathbf{X}_{\bar{S}}}[f_\tau(\mathbf{X}|\mathbf{X}_S = \mathbf{x}_S)]$ for any regression tree can be achieved by a recursive algorithm [4] with running time $O(L2^M)$, where $L$ is the number of leaves, see Algorithm 1. This requires the estimated probabilities of ending at a particular node $j$ given previous information from the ancestor nodes. If the feature used for splitting at a particular node $j$ is not used in any of the ancestor nodes of node $j$, an estimate such as in Eq (9) can be used. If the feature is used for splitting in any of the ancestor nodes, this must be accounted for by restricting to the interval of possible values the particular feature can take at node $j$.

**Algorithm 1**. Recursive algorithm for computation of $E_{\mathbf{X}_{\bar{S}}}[f_\tau(\mathbf{X}|\mathbf{X}_S = \mathbf{x}_S)]$.

```
1: Input: Tree f_τ with depth d, leaf values v = (v_1,…,v_2^d), feature used
   for splitting f = (f_1,…,f_{2^d−1}) and corresponding splitting points
   t = (t_1,…,t_{2^d−1}). Estimated probabilities of ending at a node j given pre-
   vious information, for all nodes in the tree, p = (p_1,…,p_{2^d−1}), by using
   some data (x_1, y_1), …, (x_N, y_N) of size N. The subset of features S
   with corresponding known values x_S. The left and right descendant node
   for each internal node l = (l_1,…,l_{2^d−1}) and r = (r_1,…,r_{2^d−1}). The index of a
   node j in the tree f_τ.
2: Function CondExpTree(j, f_τ, v, t, f, l, r, p)
3: if IsLeaf(j) then
4:    return v_j
5: else
6:    if f_j ∈ S then
7:      if x_j ≤ t_j then
8:        return CondExpTree(l_j, f_τ, v, t, f, l, r, p)
9:      else
10:        return CondExpTree(r_j, f_τ, v, t, f, l, r, p)
11:     end if
12:   else
13:     return CondExpTree(l_j, f_τ, v, t, f, l, r, p) p_{l_j} +
14:            CondExpTree(r_j, f_τ, v, t, f, l, r, p) p_{r_j}
15:    end if
16:  end if
17: End Function
18: CondExpTree(1, f_τ, v, t, f, l, r, p)          ▷ Start at root node.
```

**SAGE in practice.** As the expressions in Eqs (2) and (4) must be estimated, in practice we get a SAGE estimator rather than a SAGE value. However, since the SAGE estimator requires summing over all $2^{M-1}$ subsets $S \subseteq \mathcal{M}\setminus\{k\}$, for *each* feature, computing the SAGE estimator for observed data with many features becomes infeasible. In [17], the SAGE estimate is approximated through a Monte Carlo simulation process. Specifically, instead of iterating over all $2^{M-1}$ subsets, a subset $S$ is randomly sampled with replacement in each iteration out of $I$ iterations in total. The differences $w_{\mathbf{X},Y,\hat{y}}(S \cup \{k\}) - w_{\mathbf{X},Y,\hat{y}}(S)$ for each $S$ are estimated by sampling data instances with replacement and computing sample means (see [17], S1 File for details). For an arbitrarily large data set, the authors show convergence to the true SAGE estimate as $I \to \infty$. Among other things, both the accuracy and convergence speed of the algorithm naturally

depend on the number of features in the prediction model. Notice that [17] provides the degree of convergence of the approximation of the estimate, not the uncertainty in the estimate.

Keeping in mind that the SAGE *estimator* is a random variable, we argue that its uncertainty is equally important as the estimate itself. No calculation of this inherent uncertainty exists in the SAGE package nor the literature. To this end, we introduce *Sub-SAGE*, which is inspired by the SAGE framework, but consists of a reduced number of subsets $\mathcal{S} \in \mathcal{Q}$. While applicable to any number of features, it is best suited for interpreting a small number of features, or a small subset of features in a large feature set.

**Sub-SAGE.** Given hundreds or thousands of features in a model, the computation time required to get a satisfactory accurate estimate of SAGE [17] for each feature, quickly becomes unacceptable. A hybrid approach is to select a reduced subset of features of particular interest to investigate. For instance in a GWAS, such a filtering process can be achieved via a generalized linear mixed model [24], and rank importance based on the computed *p*-values. An alternative filtering procedure that also accounts for non-linear effects is described in [25]. The association between the reduced subset of promising features, and the response can then be more thoroughly investigated via a non-linear machine learning model together with SHAP values. See [25], Figure 10 for an example. To infer whether the model-based importance of the features investigated via SHAP values is also reflected in the underlying data generating process, one can compute SAGE values. However, we typically want the reduced subset of features to be sufficiently large in order to reduce the chance of missing out on important features. Even in this case, computation of SAGE values may be impractical or even infeasible. For this purpose, we introduce *Sub-SAGE*, where only a selection of the in total $2^{M-1}$ subsets are involved in the computation for each feature.

If we want to measure the importance of a feature $k$ based on its marginal effect, as well as potential pairwise interactions it may be involved in, computing $\mathcal{S} = \{\emptyset\}$ and $\mathcal{S} = \{m\}$ for $m = 1, \ldots, k-1, k+1, \ldots, M$ is sufficient. In addition, by including $\mathcal{S} = \{1, \ldots, k-1, k+1, \ldots, M\}$, the set of all features except feature $k$, can be used to measure the importance of feature $k$ in the presence of all features at the same time.

**Definition 4.** Let $\mathcal{Q}_k$ denote the set of subsets including $\mathcal{S} = \{\emptyset\}$, $\mathcal{S} = \{m\}$ for $m = 1, \ldots, k-1, k+1, \ldots, M$, and $\mathcal{S} = \{1, \ldots, k-1, k+1, \ldots, M\}$. We define the Sub-SAGE value, $\psi_k$, for feature $k$ as

$$\psi_k(\mathbf{X}, Y, \hat{y}) = \sum_{\mathcal{S} \in \mathcal{Q}_k} \frac{|\mathcal{S}|!(M - |\mathcal{S}| - 1)!}{3(M-1)!} \left[ w_{\mathbf{X}, Y, \hat{y}}(\mathcal{S} \cup \{k\}) - w_{\mathbf{X}, Y, \hat{y}}(\mathcal{S}) \right], \qquad (10)$$

Each subset is weighted such that the sum of the weights of all subsets with equal size is the same for each subset size. In addition, the sum of all weights is equal to one. Hence, the construction is similar to the weights defined for Shapley values. See S1 File for details. In this particular case, there are three possible subset sizes, and so the sum of the weights for each subset size is $\frac{1}{3}$. Shapley properties such as symmetry, dummy property and monotonicity still hold for Sub-SAGE. However, as the sum is not over all possible subsets, the Sub-SAGE values do no longer satisfy the efficiency axiom of the Shapley decomposition, which SHAP and SAGE do (see S1 File). However, we still consider the Sub-SAGE to be informative with respect to feature importance via the computed differences $w_{\mathbf{X}, Y, \hat{y}}(\mathcal{S} \cup \{k\}) - w_{\mathbf{X}, Y, \hat{y}}(\mathcal{S})$. In addition, the purpose is only to evaluate a small fraction of all features, not all of them. By only considering a reduced number of subsets $\mathcal{S}$, compared to SAGE, and only considering a reduced number of features to evaluate, both computing the Sub-SAGE estimate as well as the uncertainty in the corresponding Sub-SAGE estimator become feasible for black-box models, such as tree ensemble models and neural networks.

**Using Sub-SAGE to infer true relationships in the data.** As the goal is to infer feature importance from a black-box model using Sub-SAGE values, similar to calculating $p$-values without taking into account the effect of model selection, we must be extra careful. Any model selection procedure using training data is likely to overfit, resulting in a model consisting of possibly false relationships that are not a general property of the population from which the data was sampled. It is therefore essential that the Sub-SAGE value is calculated using independent data the model was not fitted on. We denote such independent data as test data, $(\mathbf{X}_1^0, Y_1^0), \ldots, (\mathbf{X}_{N_I}^0, Y_{N_I}^0)$, with $N_I$ samples in total. The following observation is proven S1 File.

**Observation: Sub-SAGE value in multiple linear regression.** Consider a fitted linear regression model $\hat{y}_i = \hat{\boldsymbol{\beta}}^T \boldsymbol{x}_i$. By using test data independent of the data used for constructing the linear regression model, and using the squared error loss, one can show that for a feature $k$, and any $\mathcal{S} \in \mathcal{Q}_k$:

$$w_{\mathbf{X},Y,\hat{y}}(\mathcal{S} \cup \{k\}) - w_{\mathbf{X},Y,\hat{y}}(\mathcal{S}) = 2\hat{\beta}_k \mathrm{Cov}(Y, X_k) - \hat{\beta}_k^2 \mathrm{Var}(X_k). \tag{11}$$

As the expression is independent of the subset $\mathcal{S}$, this is also equal to the Sub-SAGE value of feature $k$.

The first term in Eq (11) can be interpreted as the extent to which the influence of feature $k$ based on the model constructed using training data, is reflected in the independent test data. If the signs of $\hat{\beta}_k$ and $\mathrm{Cov}(Y, X_k)$ are identical, the first term is positive. If they differ, the Sub-SAGE value will be negative since the second term in Eq (11) is always negative. The second term $\hat{\beta}_k^2 \mathrm{Var}(X_k)$ is equal to the increased variance in the model by including feature $k$. If the model regards the feature as important (resulting in non-zero $\hat{\beta}_k$), and even if the covariance between $X_k$ and $Y$ has the same sign as $\hat{\beta}_k$, the benefit of including feature $k$ in the model will depend on the increased variance of the model. This is by construction in line with the bias-variance trade-off [26].

**Sub-SAGE applied on tree ensemble models.** SHAP values can be estimated efficiently for tree ensemble models, even with hundreds of thousands of features [25], by improving Algorithm 1 to get a significantly reduced running time of $O(TLD^2)$, for $T$ trees each of tree depth $D$ [4]. Unfortunately, there is no similar way to reduce the running time for estimation of SAGE values, as well as Sub-SAGE values, for tree ensemble models with non-linear choices of loss functions [4].

We consider a tree ensemble model consisting of $T$ trees. Consider a particular feature $k$ to compute the Sub-SAGE value as well as a subset $\mathcal{S} \in \mathcal{Q}_k$. We separate the trees in the model into two groups $\tau_k$ and the complement group $(\overline{\tau}_k)$ where $\tau_k$ is the set of trees including feature $k$ used at least once for splitting.

**Regression with squared error loss.** For regression a common loss function is the squared error between the response and prediction per sample, i.e. $\ell = (y(\mathbf{x}) - \hat{y}(\mathbf{x}))^2$. Then one can show that (see S1 File for the derivation),

$$\begin{aligned}
&w_{\mathbf{X},Y,\hat{y}}(\mathcal{S} \cup \{k\}) - w_{\mathbf{X},Y,\hat{y}}(\mathcal{S}) \\
&= E_{\mathbf{X},Y}\left[ (Y(\mathbf{X}) - V_{\mathbf{X},\hat{y}}(\mathcal{S}))^2 \right] - E_{\mathbf{X},Y}[(Y(\mathbf{X}) - V_{\mathbf{X},\hat{y}}(\mathcal{S} \cup \{k\}))^2] \\
&= E_{\mathbf{X},Y}\left[ 2Y(\mathbf{X})\left( \sum_{j \in \tau_k} V_{\mathbf{X},f_j}(\mathcal{S} \cup \{k\}) - V_{\mathbf{X},f_j}(\mathcal{S}) \right) + \left( \sum_{j \in \tau_k} V_{\mathbf{X},f_j}(\mathcal{S}) \right)^2 \right. \\
&\quad \left. - \left( \sum_{j \in \tau_k} V_{\mathbf{X},f_j}(\mathcal{S} \cup \{k\}) \right)^2 + 2\left( \sum_{j \notin \tau_k} V_{\mathbf{X},f_j}(\mathcal{S}) \right)\left( \sum_{j \in \tau_k} V_{\mathbf{X},f_j}(\mathcal{S} \cup \{k\}) - V_{\mathbf{X},f_j}(\mathcal{S}) \right) \right].
\end{aligned} \tag{12}$$

**Classification with binary cross-entropy loss.** A commonly used loss function for binary classification problems is binary cross-entropy, $\ell = -y(\mathbf{x}) \log \hat{y}(\mathbf{x}) - (1 - y(\mathbf{x})) \log (1 - \hat{y}(\mathbf{x})) = (1 - y(\mathbf{x})) \sum_{j=1}^{T} f_j(\mathbf{x}) + \log \left( 1 + e^{-\sum_{j=1}^{T} f_j(\mathbf{x})} \right)$. For this loss function, one can show that (see S1 File)

$$w_{\mathbf{X},Y,\hat{y}}(\mathcal{S} \cup \{k\}) - w_{\mathbf{X},Y,\hat{y}}(\mathcal{S}) \tag{13}$$

$$= E_{\mathbf{X},Y} \left[ (1 - Y(\mathbf{X})) \sum_{j=1}^{T} V_{\mathbf{X},f_j}(\mathcal{S}) + \log \left( 1 + \exp \left( -\sum_{j=1}^{T} V_{\mathbf{X},f_j}(\mathcal{S}) \right) \right) \right] \tag{14}$$

$$-E_{\mathbf{X},Y} \left[ (1 - Y(\mathbf{X})) \sum_{j=1}^{T} V_{\mathbf{X},f_j}(\mathcal{S} \cup \{k\}) + \log \left( 1 + \exp \left( -\sum_{j=1}^{T} V_{\mathbf{X},f_j}(\mathcal{S} \cup \{k\}) \right) \right) \right] \tag{15}$$

$$= E_{\mathbf{X},Y} \left[ (1 - Y(\mathbf{X})) \left( \sum_{j \in \tau_k} V_{\mathbf{X},f_j}(\mathcal{S}) - V_{\mathbf{X},f_j}(\mathcal{S} \cup \{k\}) \right) \right. \tag{16}$$

$$\left. + \log \left( \frac{1 + \exp \left( -\sum_{j \in \tau_k} V_{\mathbf{X},f_j}(\mathcal{S}) - \sum_{j \notin \tau_k} V_{\mathbf{X},f_j}(\mathcal{S}) \right)}{1 + \exp \left( -\sum_{j \in \tau_k} V_{\mathbf{X},f_j}(\mathcal{S} \cup \{k\}) - \sum_{j \notin \tau_k} V_{\mathbf{X},f_j}(\mathcal{S} \cup \{k\}) \right)} \right) \right]. \tag{17}$$

**Plug-in estimates.** As discussed earlier, the expression $w_{\mathbf{X},Y,\hat{y}}(\mathcal{S} \cup \{k\}) - w_{\mathbf{X},Y,\hat{y}}(\mathcal{S})$ needs to be estimated for each $\mathcal{S} \in \mathcal{Q}_k$, *and* based on data, $(\mathbf{x}_1^0, y_1^0), \ldots, (\mathbf{x}_{N_I}^0, y_{N_I}^0)$, never used during training of the model. Let $\hat{v}_{\mathbf{x}^0, y^0, f_\tau}(\mathcal{S})$ for a particular observation $(\mathbf{x}^0, y^0)$ and regression tree $f_\tau$ denote the estimate of $v_{\mathbf{x}^0, f_\tau}(\mathcal{S}) = E_{\mathbf{X}_{\overline{\mathcal{S}}}}[f_\tau(\mathbf{X}^0 | \mathbf{X}_{\mathcal{S}}^0 = \mathbf{x}_{\mathcal{S}}^0)]$ as described in Algorithm 1. A plug-in *estimate* of $\psi_k$, denoted $\hat{\psi}_k$, for a regression problem with continuous response for a tree ensemble model using the squared error loss is given by

$$\hat{\psi}_k = \sum_{\mathcal{S} \in \mathcal{Q}} \frac{|\mathcal{S}|!(M - |\mathcal{S}| - 1)!}{3(M-1)!} \left[ \frac{2}{N_I} \sum_{i=1}^{N_I} y_i^0 \left( \sum_{j \in \tau_k} \hat{v}_{\mathbf{x}_i^0, f_j}(\mathcal{S} \cup \{k\}) - \hat{v}_{\mathbf{x}_i^0, f_j}(\mathcal{S}) \right) \right.$$

$$+ \frac{1}{N_I} \sum_{i=1}^{N_I} (\sum_{j \in \tau_k} \hat{v}_{\mathbf{x}_i^0, f_j}(\mathcal{S}))^2 - \frac{1}{N_I} \sum_{i=1}^{N_I} (\sum_{j \in \tau_k} \hat{v}_{\mathbf{x}_i^0, f_j}(\mathcal{S} \cup \{k\}))^2 \tag{18}$$

$$\left. + \frac{2}{N_I} \sum_{i=1}^{N_I} \left( \sum_{j \notin \tau_k} \hat{v}_{\mathbf{x}_i^0, f_j}(\mathcal{S}) \right) \left( \sum_{j \in \tau_k} \hat{v}_{\mathbf{x}_i^0, f_j}(\mathcal{S} \cup \{k\}) - \hat{v}_{\mathbf{x}_i^0, f_j}(\mathcal{S}) \right) \right].$$

The corresponding plug-in estimate for the binary cross-entropy loss given in Eq (17) can be found in a similar fashion, basically by estimating expected values as their corresponding sample means. For tree ensemble models with tree stumps (maximum depth of one for each tree), the estimate in (18) is further reduced and can be expressed as sample variance and covariance terms, see S1 File.

**Inference of Sub-SAGE via bootstrapping.** The importance of any feature may be evaluated by estimating Sub-SAGE values. Similar to SAGE, a positive Sub-SAGE value for a feature $k$ indicates that including the feature in the model is expected, based on the subsets $\mathcal{S} \in \mathcal{Q}_k$, to

reduce the loss function. However, the corresponding Sub-SAGE plug-in *estimator* given the data generating process $(\mathbf{X}_1^0, Y_1^0), \ldots, (\mathbf{X}_{N_I}^0, Y_{N_I}^0)$ from some unknown probability distribution includes uncertainty, and this should be evaluated before making any assumptions about feature importance. The complexity of the Sub-SAGE plug-in estimators makes nonparametric bootstrapping a tempting approach. One possible procedure is to iteratively, given independent data points $(\mathbf{x}_1^0, y_1^0), \ldots, (\mathbf{x}_{N_I}^0, y_{N_I}^0)$, resample the data points *with replacement* to get a bootstrapped sample $(\mathbf{x}_1^*, y_1^*), \ldots, (\mathbf{x}_{N_I}^*, y_{N_I}^*)$, and train a model for each bootstrapped sample (with potential hyperparameters fixed). For each such generated model, a corresponding plug-in estimate, $\hat{\psi}_b^*$, can be computed, and after $B$ iterations, the sample $(\hat{\psi}_1^*, \ldots, \hat{\psi}_B^*)$ can approximate $B$ realizations arising from the true distribution of the plug-in estimator. However, in a high-dimensional setting, generating even one model may be time-consuming, and there may be circumstances where only one particular model is available for the user together with a test data sample of insufficient size to train additional models. Another option is to leave the model fixed, and only bootstrap the data repeatedly to get the sample $(\hat{\psi}_1^*, \ldots, \hat{\psi}_B^*)$. In this paper, we focus on the latter procedure. A $(1 - 2\alpha)100\%$ confidence interval can be approximated by the *percentile interval* given by $[\hat{\psi}^{*(\alpha)}, \hat{\psi}^{*(1-\alpha)}]$, where $\hat{\psi}^{*(\alpha)}$ is the $100\alpha$ empirical percentile, meaning the $B \cdot \alpha$th least value in the ordered list of the samples $(\hat{\psi}_1^*, \ldots, \hat{\psi}_B^*)$. The accuracy in the percentile interval increases for larger number of bootstrap samples. The algorithm of the paired bootstrap applied specifically to tree ensemble models is given in Algorithm 2. Notice that for each bootstrap sample, the probability estimates in the trees need to be updated according to Eq (9). In situations where the plug-in estimator is biased, or there is skewness in the corresponding distribution, the bias-corrected and accelerated bootstrap [27], may give even more accurate confidence intervals at the cost of considerable increase in computational efforts.

**Algorithm 2** Paired bootstrap of Sub-SAGE value with percentile interval

```
1: Given independent test data (x₁⁰,y₁⁰),...,(x_{N_I}⁰,y_{N_I}⁰), model ŷ(x) = ∑_{τ=1}^{T} f_τ(x),
   feature k, a loss function and α to estimate (1 - 2α)100% confidence
   interval:
2: Pre-allocate vector BootVec of length B, the total number of boot-
   strap samples.
3: for b = 1, 2, ..., B do
4:   Resample data N_I times with replacement to get
5:   (x₁*,y₁*),...,(x_{N_I}*,y_{N_I}*)
6:   Update probabilities estimates in all the trees in ŷ(x) to get p*
7:   BootVec[b] = ψ̂_k*
8: end for
9: Percentile interval given by [ψ̂^{*(α)},ψ̂^{*(1-α)}]
```

## Proof of concept—With known underlying data generating process

In this section, we exemplify the Sub-SAGE method on synthetic data with a known relationship defined as

$$
\begin{aligned}
f(\mathbf{X}_i) \quad &= a_0 + a_1 X_{i,1} + a_2 X_{i,2} + a_{21} X_{i,1} e^{X_{i,2}} + a_3 X_{i,3}^2 + a_4 \sin(X_{i,4}) \\
&\quad + a_5 \log(1 + X_{i,5}) - X_{i,5} I(X_{i,6} > 7) + \epsilon_i,
\end{aligned}
\tag{19}
$$

with $a_0 = -0.5$, $a_1 = 0.03$, $a_2 = -0.05$, $a_{21} = 0.3$, $a_3 = 0.02$, $a_4 = 0.35$, $a_5 = -0.2$, and where the

features are sampled from the following distributions

$$X_1 \sim \text{Binom}(\text{size} = 2, p = 0.4)$$

$$X_2 \sim \text{Binom}(\text{size} = 2, p = 0.04)$$

$$X_3 \sim \Gamma(\text{shape} = 10, \text{rate} = 2)$$

$$X_4 \sim \text{Unif}(0, \pi) \tag{20}$$

$$X_5 \sim \text{Poisson}(\lambda = 15)$$

$$X_6 \sim \text{N}(\mu = 0, \sigma = 10)$$

$$\epsilon \sim \text{N}(\mu = 0, \sigma = 2).$$

In addition, we generate 94 noise variables: $j = 7, \ldots, 47$ with a normal distribution $X_j \sim \text{N}(\mu_j, \sigma_j)$ and $j = 48, \ldots, 100$ with a binomial distribution $X_j \sim \text{Binom}(2, p_j)$ where $\mu_j$, $\sigma_j$ and $p_j$ are sampled from a uniform distribution. In realistic applications, the data distributions and relations are unknown and the purpose of model fitting is to estimate the relations between variables. Data is generated to give a total of 16000 samples, and then separated randomly in three disjoint subsets: Data for training (50%), data for evaluation during training (30%) and independent test data (20%) used for estimating Sub-SAGE values. We fit an ensemble tree model using XGBoost [28] to the true influential features 1, ..., 6 together with the noise variables 7, ..., 100.

The hyperparameters are fixed to max_depth = 2, learning rate $\eta = 0.05$, subsample = 0.7, regularization parameters $\lambda = 1$, $\gamma = 0$ and colsample_bytree = 0.8 with early_stopping_rounds = 20 using training data ($n = 8000$) and validation data ($n = 4800$), and a squared error loss. See [28] for details about the hyperparameters. This results in a final model including a total of 230 trees and 62 unique features out of the 100 input-features.

From the trained model, each feature is given a score to evaluate its feature importance *based on the model*. We apply the expected relative feature contribution (ERFC) [25], given data of size $N$, which is basically a summary score from the corresponding SHAP values for each feature and individual data point,

$$\kappa_k = \sum_{i=1}^{N} \frac{|\phi_{i,k}^{\text{SHAP}}(\mathbf{x}_i, \hat{y})|}{|\phi_0^{\text{SHAP}}| + \sum_{j=1}^{K} |\phi_{i,j}^{\text{SHAP}}(\mathbf{x}_i, \hat{y})|}, \tag{21}$$

with $\phi_0^{\text{SHAP}} = v_{\mathbf{x}, \hat{y}}(\emptyset)$. The ERFCs scores can be computed based on the data used to construct the model, as we only need to measure what the model considers important. The features with the largest ERFC-values are then considered the most promising ones *based on the model*. Depending on your hypothesis of interest, one can evaluate the uncertainty in the feature importance by computing Sub-SAGE estimates with corresponding bootstrap-derived percentile intervals. However, it is important that the Sub-SAGE estimates are calculated based on independent test data never used during training.

From the trained model, we compute the ERFC based on the training data and validation data together ($n = 12800$), and Table 1 shows the top 10 features with the largest ERFC-values.

We see that the XGBoost model has accurately ranked the most influential features among the top 10 list, for this rather simple relationship. These scores, based on SHAP values, are only with respect to what the *model* considers important. The Sub-SAGE can now be applied to infer whether feature importance from the model is also reflected in the data. As an example, let us consider features 6, 1, 2 and 12 where feature 6 has a strong influence,

**Table 1. The resulting ranking based on the expected relative feature contribution (ERFC) after having trained an XGBoost model consisting of six influential features ($x_1 - x_6$) and 94 noise features ($x_7 - x_{100}$).**

| Feature | ERFC |
|---|---|
| $x_6$ | 0.48 |
| $x_5$ | 0.060 |
| $x_3$ | 0.026 |
| $x_1$ | 0.022 |
| $x_4$ | 0.0036 |
| $x_2$ | 0.0030 |
| $x_{12}$ | 0.0028 |
| $x_{30}$ | 0.0022 |
| $x_{40}$ | 0.0019 |

feature 1 has a weaker influence, and feature 2 has the weakest influence, while feature 12 has no influence with respect to $f(\mathbf{x}_i)$ in Eq (19). Their Sub-SAGE estimate along with histograms to estimate the corresponding distribution of the Sub-SAGE estimators, using $B = 1000$ bootstrap samples, are shown in Fig 2 for training plus validation data as well as for independent test data.

We see that Sub-SAGE values inferred using training data overestimate the false influence of feature 12, while using the test data correctly indicates that feature 12 has a weak or no influence. We also see from the other histograms that using the training data underestimates the uncertainty in the Sub-SAGE estimate. By using the test data for computation of the Sub-SAGE estimates, the estimated 95% percentile intervals of the Sub-SAGE values for each feature are 6 : (39.45, 44.15), 1 : (−0.038, 0.14), 2 : (−0.043, 0.040) and 12 : (−0.030, 0.0050). These ranges allow us to conclude that feature 6, correctly, is highly influential, while feature 12 is highly unlikely to have any influence. Moreover, feature 1 is likely to be influential, while the influence of feature 2 is very uncertain.

To correct for a potential bias in the plug-in estimator of the Sub-SAGE as well as potential changes in the standard deviation of the estimator at different levels, the bias-corrected and accelerated bootstrap confidence interval may give more accurate bootstrap confidence intervals [27]. This results in the intervals 6 : (39.45, 44.13), 1 : (−0.034, 0.14), 2 : (−0.047, 0.037) and 12 : (−0.031, 0.0040), with only negligible changes from the percentile confidence intervals.As the data generating process is known, we can compare the true SHAP value at each point with the corresponding SHAP value from the fitted model. Fig 3 shows that the influence of feature 6 is quite accurately modelled, while the effect of feature 1 and particularly feature 2 is highly underestimated when $x_1 = 1$ and $x_2 = 2$. Since there is an interaction effect involving features 1 and 2, the SHAP value of feature 1 depends on the value of feature 2. It also becomes clear that feature 12, according to the model, has a negative trend in the SHAP value, but the true SHAP value is equal to zero (no importance), regardless of the value of feature 12. See S1 File for derivations. This shows an example where feature 12 is erroneously attributed high predictive importance by SHAP, while the corresponding Sub-SAGE value correctly indicates it has no importance.

We can explore the results even further by comparing the estimated Sub-SAGE values from the tree ensemble model with the exact Sub-SAGE value from the true model in (19), as well as by comparing with the exact SAGE value. The results are given in Table 2 and the details of the computations in S1 File.

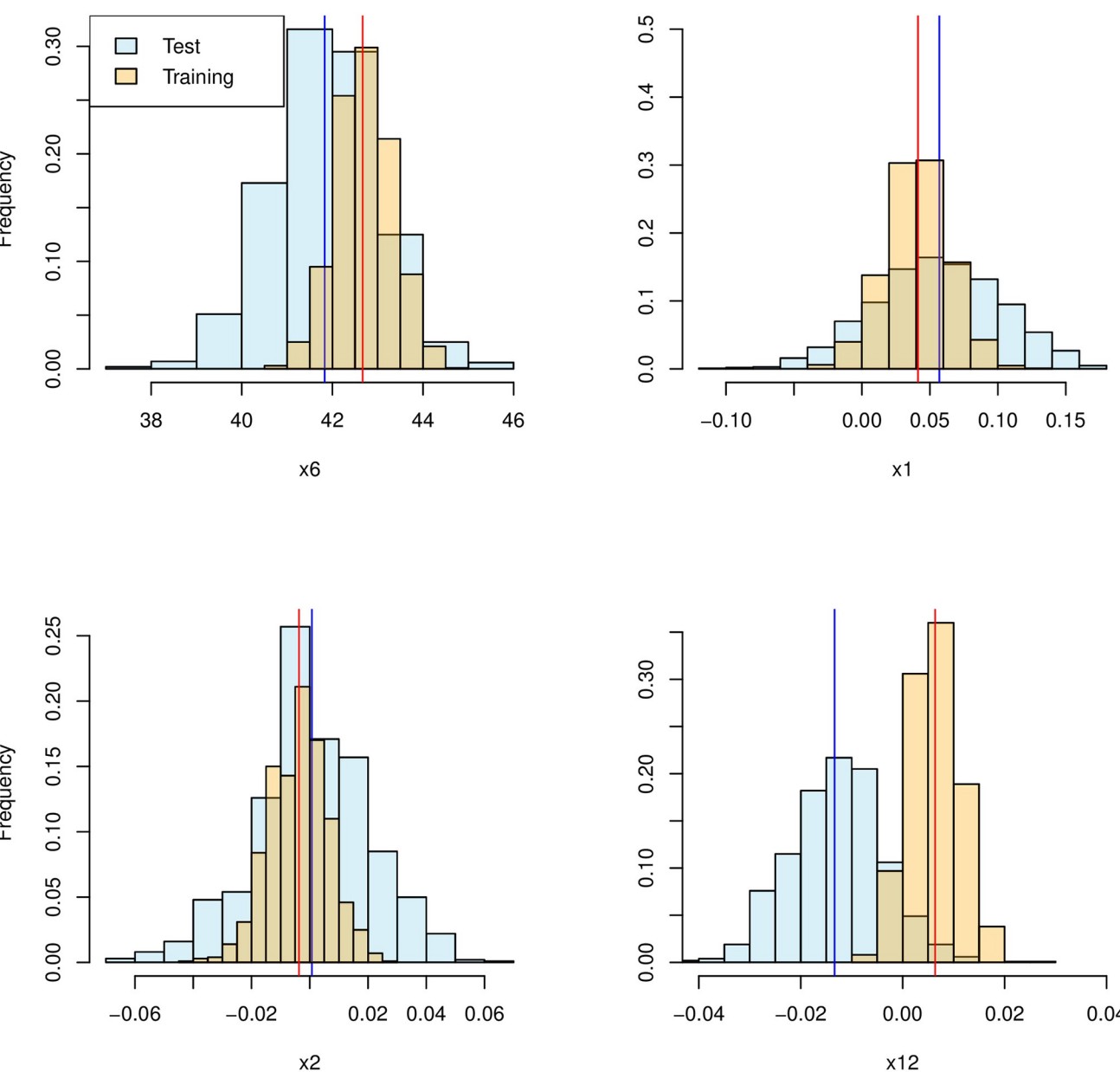

**Fig 2. The estimate of the Sub-SAGE, and the corresponding bootstrap distribution for the synthetic data for features $x_6$, $x_2$, $x_1$ and $x_{12}$, when applying data used during training (orange), and independent test data (blue).**

The comparison shows that the XGBoost model has captured almost all of the relationship between the response and feature 6, while the influence of feature 1, and particularly feature 2 has been underestimated. It also shows the small difference between the true Sub-SAGE and SAGE value with respect to the model in (19), and therefore the small gain of computing the SAGE value in this particular case. However, as the true relationship in this case is restricted to pairwise interactions, the insignificant difference between SAGE and Sub-SAGE cannot in general be anticipated for instance when the true relationship includes higher-order interactions.

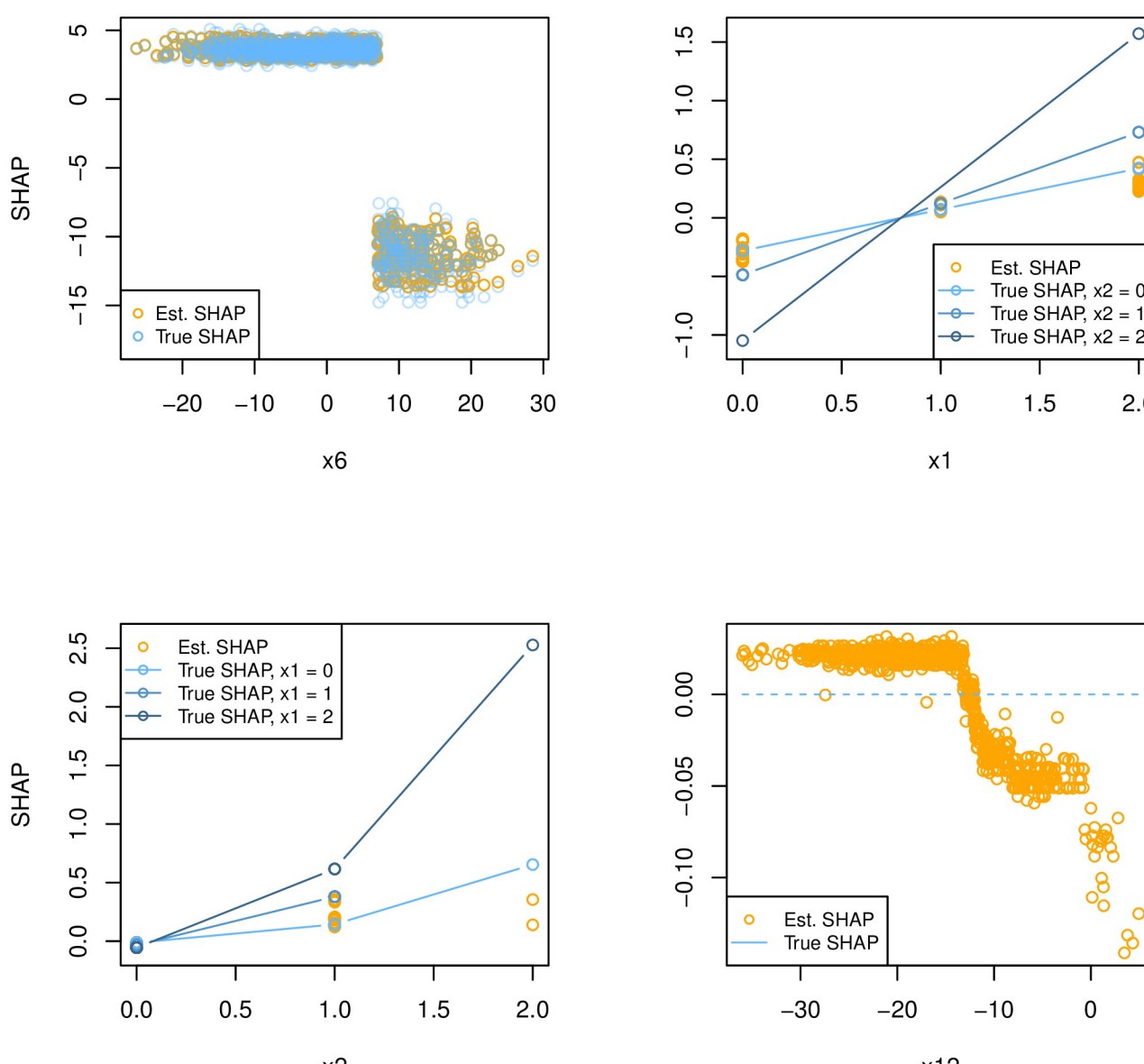

**Fig 3. Comparison of true SHAP value for each data point with the estimated SHAP value from the model fitted on the synthetic data, Eq (19). The deviations explain the reasons behind under- and overestimation of feature importance using SHAP values.**

**Table 2. Exact Sub-SAGE and SAGE value for features 1, 2, 6 and 12 using true model in (19), denoted TM, together with estimated Sub-SAGE value using the trained XGBoost model, denoted XGB, based on syntetic data based on (19).**

| Feature | Sub-SAGE (TM) | Sub-SAGE, (XGB) | SAGE (TM) |
|---------|---------------|------------------|-----------|
| 6 | 42.37 | 41.83 | 42.64 |
| 1 | 0.071 | 0.057 | 0.072 |
| 2 | 0.017 | 0.00073 | 0.018 |
| 12 | 0 | -0.0133 | 0 |

## Results

### Application on genetic data using the UK Biobank resource

To demonstrate the ability of Sub-SAGE on observed data, we consider a realistic machine learning problem using both genetic and non-genetic data of moderate size from UK Biobank.

The aim is to infer the influence of specific features with respect to obesity (BMI $\geq$ 30), by training an XGBoost model and computing Sub-SAGE values. We treat this as a classification problem between the categories obese and non-obese (see [25] for details). Of particular interest is whether any genetic markers are important. The most used method in this setting is a so-called genome-wide association study (GWAS), where each genetic variant is tested individually in a general linear (mixed-effects) regression model [24, 29]. A corresponding $p$-value of less than $5 \times 10^{-8}$ is often considered statistically significant. This tiny significance level is chosen due to the multiple comparison problem [30]. When the same association is replicated in an independent data set, the association is considered to be robust.

We study a particular XGBoost model constructed in [25]. The model was trained to predict the probability of an individual being obese, $p(Y_i = 1|\mathbf{x}_i)$, given genetic and non-genetic data, $\mathbf{x}_i$, such that

$$\text{logit}(p(Y_i = 1|\mathbf{x}_i)) = \sum_{\tau=1}^{T} f_\tau(\mathbf{x}_i),$$

consisting of $T$ regression trees. The genetic data consists of so-called *minor allele counts* or genotype values from single nucleotide polymorphisms (SNPs) filtered to limit dependence between the SNPs without significant loss of information [25]. In detail, the particular XGBoost model mentioned above was constructed after the so-called ranking process explained in [25], which ranks features by importance and filter for correlation. Using a sample of 207 015 individuals, a total of 529 024 SNPs were split into 50 randomly selected subsets, each consisting of SNPs with mutually small correlation (Pearson's correlation $r^2 < 0.2$) and unrelated individuals, with the purpose of limiting the correlation between features due to linkage disequilibrium [31], as well as reducing the effect of population stratification and cryptic relatedness [32]. For each such subset, XGBoost models in combination with cross-validation were fitted, and the importance of each SNP, taking into account all the generated models, was measured according to the model-agnostic *ERFC* score introduced in [25] based on SHAP values. The ranked list of features by importance was again filtered for correlation (Pearson's correlation $r^2 < 0.2$), and used during the so-called model fitting process based on data not used during the ranking process (see Figure 3 in [25]). The aim in the model fitting process is to find how large the portion of top-ranked features in the training data must be to get the strongest prediction model, using the PR-AUC metric [33], restricted to an ensemble model consisting of XGBoost models constructed via cross-validation (see Figure 9 in [25]). From the best-performing ensemble model, and for simplicity restricted to regression trees of maximum depth equal to two, we picked one of the XGBoost models from this ensemble model as the particular model to investigate further in this paper. Non-genetic features included during the process are sex, age, physical activity frequency, intake of saturated fate, sleep duration, stress and alcohol consumption. Not surprisingly, these non-genetic features are considered most important, and therefore also included in our particular XGBoost model to investigate. During the model fitting process, the hyperparameters were optimized within a restricted region of the hyperparameter space (given in Table 4 in [25]). When restricting to regression trees of maximum depth equal to two, the best performing ensemble model resulted in the following hyperparameter values: Learning rate $\eta = 0.05$, *colsample = subsample =*

**Table 3. The resulting ranking based on the expected relative feature contribution (ERFC) for the particular XGBoost model investigated based on training data consisting of 64 000 individuals from UK Biobank.**

| Feature | ERFC |
|---|:---:|
| Alcohol intake frequency | 0.088 |
| Genetic sex | 0.086 |
| Physical activity frequency | 0.073 |
| Intake of saturated fat | 0.044 |
| Sleep duration | 0.036 |
| Stress | 0.034 |
| Age at recruitment | 0.033 |
| rs17817449 | 0.017 |
| rs489693 | 0.012 |
| rs1488830 | 0.011 |
| rs13393304 | 0.010 |
| rs10913469 | 0.01 |
| rs2820312 | 0.0086 |

$colsample\_by\_tree = 0.8$, $max\_depth = 2$, $\lambda = 1$, $\gamma = 1$, $early\_stopping\_rounds = 20$, with binary cross-entropy loss. The particular XGBoost model to be investigated is constructed based on 64 000 individuals from UK Biobank, and includes 532 features spread along a total of 607 trees. Computing SAGE values for all features would be very time-consuming, if not infeasible. Other feature importance scores that are faster to compute, yet less trustworthy, such as SHAP or ERFC must typically be used instead when pre-evaluating the importance of each feature. The features with the largest ERFC-scores based on the training data are given in Table 3. We consider those to be the most relevant for further investigation.

Before we compute the Sub-SAGE values, we check each feature in the context of domain knowledge. While the non-genetic features are considered the most important, the most important SNP according to the model is rs17817449. This SNP is connected to the *FTO* gene at chromosome 16, and has previously been (statistically significant) associated with obesity in a large number of genome-wide association studies including different independent data sets [21]. The SNP rs13393304 at chromosome 2 has previously been associated with obesity using UK Biobank data [34]. From the PheWeb platform [35], a generalized linear mixed model [24], based on TopMed imputation on each individual [36], was constructed separately on each trait out of a total of 1419 traits in UK Biobank. In this case, the SNP rs489693 is second most associated with obesity, yet not statistically significant with $p$-value = $2.3 \times 10^{-7}$. Likewise, for the SNPs rs1488830, rs10913469 and rs2820312 the computed $p$-values are $2.2 \times 10^{-3}$, $7.1 \times 10^{-5}$ and $1.1 \times 10^{-2}$ respectively, and therefore not declared statistically significant with respect to obesity. However, the association between the SNP rs2820312 and hypertension is in fact statistically significant ($2.5 \times 10^{-9}$) in the PheWeb platform, and obesity is known to be a risk factor for hypertension [37].

The uncertainty of the feature importance of the SNPs rs17817449, rs13393304 and rs2820312 in Table 3 are explored more thoroughly by computing Sub-SAGE estimates including paired bootstrap-derived percentile intervals, with $B = 1000$ bootstrap samples, by using 20000 (unrelated White-British) participants from UK Biobank not used while training the model. We also compute Sub-SAGE for the randomly selected SNP rs7318381, which has never been associated with obesity, and with a small ERFC in the XGBoost model (0.0016). The results are given in Fig 4.

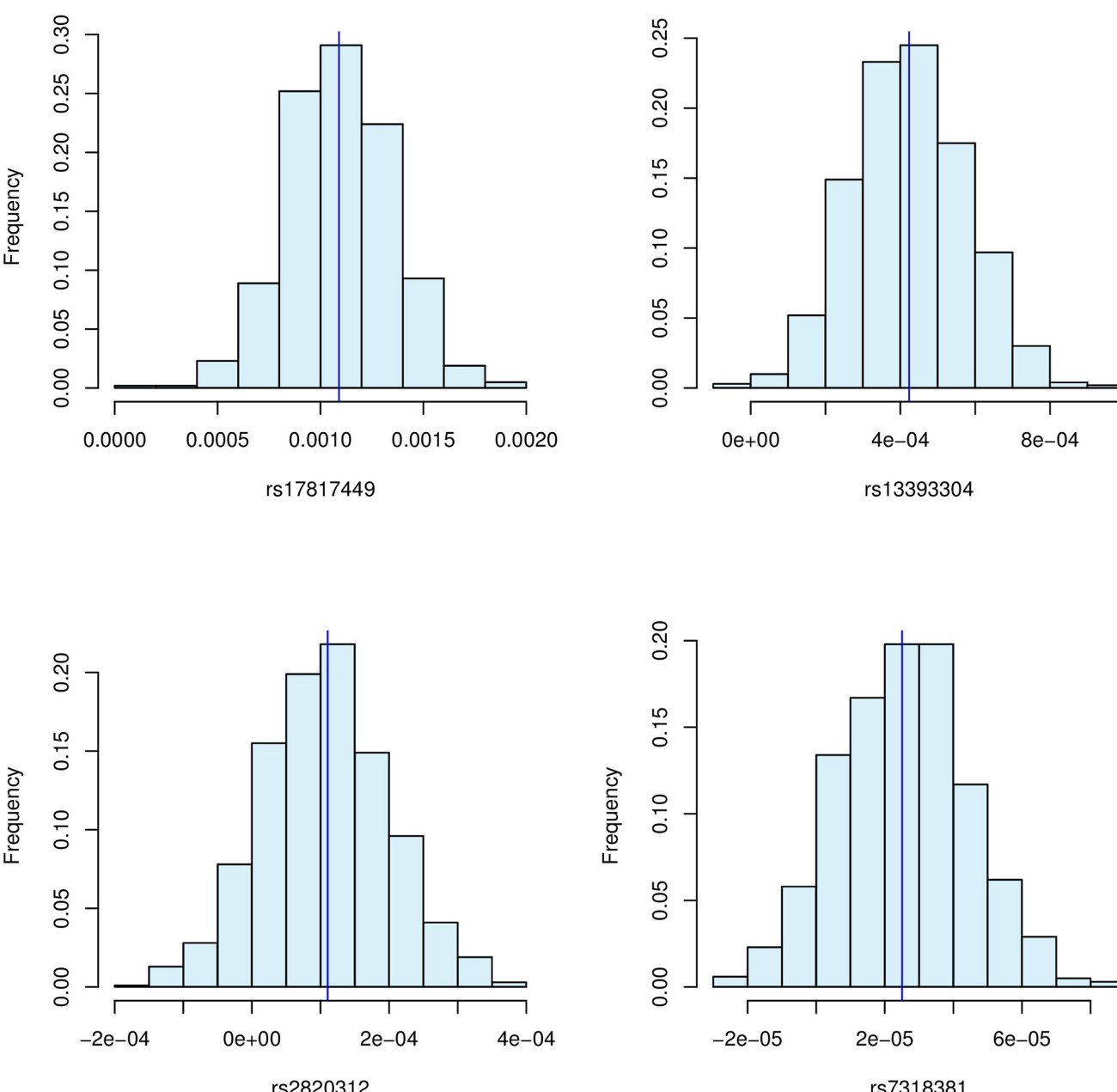

**Fig 4. The estimates and corresponding uncertainties in the Sub-SAGE values for the four SNPs agree with previous studies (GWAS) regarding SNP-association with obesity.**

The Sub-SAGE values do indicate that both rs17817449 and rs13393304 are highly likely to be associated with obesity. The 95% percentile interval of the Sub-SAGE value for rs17817449 is (0.0006, 0.0016), and (0.00014, 0.00073) for rs13393304. The SNPs rs2820312 and rs7318381 are less likely to be associated with obesity, and if they are true associations, the uncertainties in the estimates indicate that the effects are microscopic. The 95% percentile intervals for rs2820312 is $(-7.08 \times 10^{-5}, 2.95 \times 10^{-4})$, and $(-1.13 \times 10^{-5}, 6.32 \times 10^{-5})$ for rs7318381.

When dealing with relatively large data sizes such as for the genetic example above, the bias-corrected and accelerated bootstrap interval can become infeasible due to the estimation

of the acceleration parameter. However, as the acceleration parameter is proportional to the skewness of the bootstrap distribution, and if the bootstrap distribution indeed has a small skewness, as is the case here, it is often sufficient to set the acceleration parameter equal to zero. This gives no change in the percentile intervals of rs17817449 and rs13393304, but the bias-corrected 95% bootstrap intervals of rs2820312 and rs7318381 become $(-6.10 \times 10^{-5}, 3.0 \times 10^{-4})$ and $(-1.18 \times 10^{-5}, 6.19 \times 10^{-5})$ respectively. These are negligible changes, indicating that the plug-in estimates are low-biased.

With the number of individuals and size of the model explained above the provided R and Rcpp code performs a single Sub-SAGE estimate in around 15–20 minutes using CPUs available at the Farnam Cluster from Yale Center for Research Computing. The bootstrap samples were accomplished using job arrays in a high-performance computing environment, and were completed within around 1.5 hours for each feature.

## Discussion and conclusion

We present a Shapley-value-based framework for inferring the importance of individual features, including uncertainty in the estimator. We argue that SAGE values, or Sub-SAGE values, are more appropriate for quantifying global feature importance than SHAP values, as SHAP values only depend on the fitted model itself, good or bad, while SAGE and Sub-SAGE values additionally account for the performance with respect to the true data generating process via the loss function. Effectively, using SAGE and Sub-SAGE for inferring feature importance reduces the false positive rate compared to when using SHAP. As the computation of SAGE values quickly becomes challenging for increasing number of features, we introduce the Sub-SAGE value as an appropriate alternative. We demonstrate how to infer feature influence for a tree ensemble model with high-dimensional data using Sub-SAGE and paired bootstrapping. As an example, we use XGBoost, a gradient tree-boosting model, applied to both a known data generating process, as well as realistic high-dimensional data. We emphasize the importance of using test data, independent of data used to construct the model, to compute Sub-SAGE estimates.

The particular choice of $\mathcal{Q}_k$ in the definition of Sub-SAGE was based on the fact that marginal effects and pairwise interaction effects are accounted for. An alternative is to include all subsets with cardinality restricted to some value. Yet another approach is to sample a restricted number of subsets $\mathcal{S}$ following the same probability distribution as for the Kernel SHAP method, see [3] for details. The main idea is that the Shapley consistency property is not a necessity if the question is whether a feature $k$ is regarded as important with respect to a particular prediction model.

It is important to notice that the percentile intervals, constructed to evaluate the uncertainty in the Sub-SAGE estimate, themselves include uncertainty. The uncertainty of the percentile intervals depends on the number of bootstraps, $B$, as well as the size $n$ of data samples. However, in addition, the uncertainty also depends on the ratio $p/n$, where $p$ is the total number of features *used* in the model (not necessarily the number of input features for constructing the model). This fact is particularly important in high-dimensional problems, and it has been discussed for instance in [38]. When applied to linear models, one observation from a simulation is for instance that the paired bootstrap becomes more conservative (loss of power) the larger the ratio $p/n$ is. Observe that for the simulation example above, $p/n = 62/3200 = 0.019$, while for the genetic data, the ratio is $p/n = 533/20000 = 0.027$, deliberately chosen to be small in order to account for the problems arising when $p/n$ becomes too large. For the genetic data, a filtering process is first needed as the data from UK Biobank originally includes around 530000 SNPs and 207000 individuals ($p/n = 2.56$). The applied filtering method and potential pitfalls are described in [25].

It seems reasonable to apply the same loss function in the Sub-SAGE estimate as the loss function that was used to construct the model. However, there may be situations where it is meaningful to compute the Sub-SAGE values for a different loss function than the loss function used during training in order to make more objective interpretations. This may for instance be the case when the model is provided 'as is', and you do not know the training loss function, or when using adapted loss functions, e.g. weighted binary cross-entropy, but the interpretation is relevant for a standard cross-entropy.

If we use Sub-SAGE to infer importance of a feature, let the null hypothesis be that the corresponding Sub-SAGE value is less than or equal to zero. A simple procedure to investigate this is to construct a one-sided $(1 - \alpha)100\%$ percentile interval, and reject the null hypothesis if the corresponding lower bound is greater than zero. However, the use of a bootstrap confidence interval to construct a hypothesis test often has a low statistical power [39]. If in addition several features are tested simultaneously, a multiple testing procedure would be necessary in order to control the false positive rate [30]. We leave it to future research as to how to construct a more powerful hypothesis testing procedure, and how to control the false positive rate in a multiple testing procedure.

The statistical power when inferring feature importance based on Sub-SAGE will rely on the model uncertainty, the degree to which the prediction model has captured the true relationship between a particular feature and the response.

In this work we have assumed all features to be mutually statistically independent, an unrealistic scenario in most cases. If many features are statistically dependent, one is required to estimate conditional expected values (see [3] for details). Even for medium-size data sets this often becomes very tedious and even infeasible in most cases. One possibility is to use principal component analysis for dimensionality reduction, but this is not straightforward if we need the features of the model to be meaningful, and thereby explainable. In addition, principal component analysis is based on variance in the features and not explanatory power. An important line of future research to allow for evaluation of feature importance in a high-dimensional setting is dimensionality reduction that preserves interpretability.

The estimates provided by Sub-SAGE, as for SHAP values, will be more reliable the better the overall predictions from the model. Recent research has shown the strong benefit of including individual polygenic risk scores [40] as a covariate in the XGBoost model for greater performance in predictions of susceptibility for several phenotypes [41]. Computation of Sub-SAGE values in this case would depend on the correlation between PRS and individuals SNPs used in the XGBoost model, and if so the need for estimating conditional expected values.

In this paper, we have focused on the marginal effect of each feature, which is the total effect of the feature including the isolated effect of the feature as well as possible interaction effects the feature may be involved in. In principle, the construction of SHAP interaction values to quantify pairwise interaction effects, as introduced in [4] and applied in [25], could be extended to also include SAGE or Sub-SAGE interaction values. In this setting, the running time is even more computationally demanding. In that respect, the idea of reducing the number of subsets to include in the Shapley-based computations can be one approach to reduce the computation time. Interesting future research would be how to apply the idea of Sub-SAGE to search for interaction effects.

## Supporting information

**S1 File. Supporting information including proofs and derivations throughout the paper.**
(PDF)

## Acknowledgments

We thank the Yale Center for Research Computing for guidance and use of the research computing infrastructure. We thank The Gemini Center for Sepsis Research for establishing cooperation with Yale School of Public Health.

## Author Contributions

**Conceptualization:** Pål Vegard Johnsen, Inga Strümke, Mette Langaas, Andrew Thomas DeWan, Signe Riemer-Sørensen.

**Data curation:** Pål Vegard Johnsen.

**Formal analysis:** Pål Vegard Johnsen, Inga Strümke, Mette Langaas, Signe Riemer-Sørensen.

**Investigation:** Pål Vegard Johnsen, Inga Strümke, Signe Riemer-Sørensen.

**Methodology:** Pål Vegard Johnsen, Inga Strümke, Mette Langaas, Signe Riemer-Sørensen.

**Project administration:** Andrew Thomas DeWan, Signe Riemer-Sørensen.

**Resources:** Andrew Thomas DeWan.

**Software:** Pål Vegard Johnsen.

**Supervision:** Inga Strümke, Mette Langaas, Andrew Thomas DeWan, Signe Riemer-Sørensen.

**Validation:** Pål Vegard Johnsen, Inga Strümke, Mette Langaas, Signe Riemer-Sørensen.

**Visualization:** Pål Vegard Johnsen, Signe Riemer-Sørensen.

**Writing – original draft:** Pål Vegard Johnsen, Inga Strümke, Mette Langaas, Signe Riemer-Sørensen.

**Writing – review & editing:** Pål Vegard Johnsen, Inga Strümke, Mette Langaas, Signe Riemer-Sørensen.

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
