## [Decision Letter · Decision Letter 0]

4 Oct 2022

Dear Mr. Johnsen,

Thank you very much for submitting your manuscript "Inferring feature importance with uncertainties with application to high-dimensional genotype data" for consideration at PLOS Computational Biology.

As with all papers reviewed by the journal, your manuscript was reviewed by members of the editorial board and by several independent reviewers. In light of the reviews (below this email), we would like to invite the resubmission of a significantly-revised version that takes into account the reviewers' comments.

We cannot make any decision about publication until we have seen the revised manuscript and your response to the reviewers' comments. Your revised manuscript is also likely to be sent to reviewers for further evaluation.

One of the reviewers pointed out that the Github repository linked in your submission contains code for an earlier work and does not contain the subSAGE method described in your manuscript. In your revision, please advise whether an open source code implementation of your method is available to the community, as required by the journal. 

Sincerely,

Shaun Mahony

Academic Editor

PLOS Computational Biology

Ilya Ioshikhes

Section Editor

PLOS Computational Biology

Reviewer's Responses to Questions

**Comments to the Authors:**

Reviewer #1: In this work, Johnsen and colleagues build on SAGE to estimate feature importance with uncertainties. While SAGE has the advantage (compared to SHAP values) of providing a global estimate of feature importance via a loss function, it is ultimately an estimate, and as such, the authors argue, we should be interested in quantifying the uncertainty associated with this estimate. They propose sub-SAGE as an approach to do this in a computationally efficient manner, by using a reduced number of subsets/features and providing bootstrapped confidence intervals. To demonstrate their approach, they provide a proof-of-concept example using a small synthetic dataset, as well as a further example with a tree ensemble model using genotype data from the UK Biobank to infer importance for genetic and non-genetic features in an obesity phenotype. As noted by the authors themselves, there are some potential limitations to this approach, namely: using a limited number of subsets results in sub-SAGE values which no longer satisfy the efficiency axiom, the use of independent test data is required, and the accuracy of percentile confidence intervals may vary depending on number of bootstraps (although admittedly this variance may be relatively minor).

General comment:

Overall, the manuscript is well-written, providing rigorous detail on the methodological approach taken. Improving the interpretability of machine learning algorithms as applied to high-dimensional biological data (as well as more generally) is an important area of research, and understanding the uncertainty associated with SAGE estimates should prove of interest and benefit to the field.

Specific comments:

- The high-dimensional example provided is still a relatively small dataset given the current size of GWAS studies - can the authors provide an indication of run-time to estimate the sub-SAGE values?

- the authors propose selecting the reduced number of features based on either domain knowledge or by "selecting the top tier". In the case of the former, might this limit the ability for discovery of novel genetic relationships? For the latter, there is no indication as to what this entails - some further details would be helpful.

For the UKBiobank example:

- Have the XGBoost hyperparameters mentioned been optimised?

- Is there any attempt to account for correlation in SNP importance, based on e.g. LD structure and if not, how does this affect importance estimates?

- Recently published work (https://www.nature.com/articles/s42003-022-03812-z) has shown the benefit of including both non-linear contributions and interactions between SNPS (ascertained via XGBoost) as well as simple additive PRS when explaining heritability in complex phenotypes - do the authors see any potential benefit to such a combined approach with sub-SAGE estimates?

Reviewer #2: The authors introduce a novel method for feature selection. Although the method and concept is interesting, the methodology and application of this method bears many limitations as outlined below:

The results from synthetic data are not very convincing that the method identifies the most important variables with high confidence. The interpretation of score is difficult and no understanding of the range of the score is provided. The testing dataset helps understand the confidence a bit but in real world identifying a testing data every-time is a big task.

The simulations are only performed on a sample size of 16,000 which is a moderate samples size but would not be considered large dataset. Authors should show power, sensitivity and specificity of the method on a large set of simulations to represent real world datasets. Variables for simulations should include combinations of:

Sample sizes

Minor allele frequencies

Noise to Signal Ratio

Penetrance and prevalence of the disease

Real world data application: UKBB dataset consists of >400K samples, however authors only used 64000 samples. It is unclear why they chose only this subset. If the point of the method is to perform GWAS, then the new proposed method should be able to run GWAS on as large of dataset as the conventional methods. It would be imperative to run the analyses on all samples and all variants (possibly an imputed GWAS dataset).

Non-genetic features such as age, sex, ancestry are among the most important features in the real world analyses. However, no interpretation is provided on the significance of known SNPs while adjusting for these confounders. A standard GWAS analyses would include adjusting for any covariates, how does this method account for covariates?

Standard GWAS methods assume an additive model and are most powered at identifying additive effects. An improvement over existing GWAS methods would be to identify non-additive effects. Is this method able to identify only known signals or any novel signals as well for BMI that might help elucidate the genetic underpinnings of obesity more than standard GWAS fro UKBB? The previous method published in BMC Bioinformatics focussed on GXG and GXE, does this method also identify those effects, unclear from the manuscript?

Reviewer #3: Johnsen et al. consider the problem of computing global feature importance in high-dimensional datasets. Prior work on computing global feature importance such as SAGE do not scale with the number of features (even when using Monte Carlo approximations). This work proposes a modified measure of feature importance, sub-SAGE, that only considers the importance of a feature based on its marginal and pairwise interactions. This definition has the advantage of enabling efficient computation especially when used to explain tree-based models. Further, the authors propose the use of Bootstrap resampling to compute the uncertainty of the sub-SAGE estimates. While there has been increasing attention paid to estimating feature importance, the notion of uncertainty in these estimates has not been adequately studied (beyond the notion of uncertainty due to Monte Carlo sampling). This is the first work that attempts to provide uncertainty estimates to their estimates of sub-Sage values which is an important step to interpret and trust feature importance measures. Nevertheless, there are several major comments that the

Limitations:

Major:

1. The assumption of feature independence is quite strong, especially if the target application is genetic data, as the authors also pointed out. The authors need to better justify the target application given the assumptions underlying their method. When the authors consider SNPs that are associated with obesity in the UK Biobank, many of these SNPs are likely to be highly correlated due to linkage disequilibrium. Do the authors prune SNPs that are in strong LD?

2. The title refers to high-dimensional genotype data which might suggest that these methods are being applied in the setting where the number of SNPs exceeds the number of individuals/samples. However this is not the case from the UK Biobank application (< 3,000 SNPs and 64,000 samples). The authors might want to reword the title to be more accurate.

3. The ability to scale the algorithm to high-dimensional data relies on considering the pair-wise feature interaction and marginal feature contribution in sub-SAGE. While this is a reasonable choice, a systematic analysis of the loss of power due to such an approximation is missing. The comparison of sub-SAGE and SAGE in Table 2 is done under a simulation that assumes only pair-wise feature interactions.

4. One contribution the paper claims is the development of an efficient algorithm for global feature importance uncertainty estimation under the tree structure. An alternative approach to achieve the same goal is to instead estimate instance-wise feature attribution values using Tree-SHAP, approximate the global feature importance using the expectation of individual feature attribution estimates, and then use the bootstrap similarly to estimate uncertainty. (Covert et al. have justified the equivalency between these two approaches). The authors should clearly state the advantage of computing global feature importance directly compared to calculating it with the aggregated instance-wise feature importance.

5. A more general question for the authors is to demonstrate the usefulness of the uncertainty estimates. One natural application is to perform feature selection. If the intention of the uncertainty estimation is for feature selection purposes, it would be valuable to compare it with some commonly used feature selection techniques in the area. Additional analyses and discussion of the utility of uncertainty estimates would strengthen the manuscript.

Minor:

1. What does the notation y(x) in Equation 5 refer to?

2. There is a sign missing in front of the second line of Equation (19).

**Have the authors made all data and (if applicable) computational code underlying the findings in their manuscript fully available?**

Reviewer #1: Yes

Reviewer #2: **No: **The link leads to GitHub page for previously published method in BMC Bioinformatics.

Reviewer #3: Yes

PLOS authors have the option to publish the peer review history of their article (what does this mean?). If published, this will include your full peer review and any attached files.

Reviewer #1: No

Reviewer #2: No

Reviewer #3: No
---

## [Decision Letter · Decision Letter 1]

2 Jan 2023

Dear Mr. Johnsen,

Thank you very much for submitting your manuscript "Inferring feature importance with uncertainties with application to large genotype data" for consideration at PLOS Computational Biology. As with all papers reviewed by the journal, your manuscript was reviewed by members of the editorial board and by several independent reviewers. The reviewers appreciated the attention to an important topic. Based on the reviews, we are likely to accept this manuscript for publication, providing that you modify the manuscript according to the review recommendations.

Sincerely,

Shaun Mahony

Academic Editor

PLOS Computational Biology

Ilya Ioshikhes

Section Editor

PLOS Computational Biology

Reviewer's Responses to Questions

**Comments to the Authors:**

Reviewer #1: I thank the authors for their detailed response to comments and for the additional text added to the manuscript.

Reviewer #3: In the revised version, the authors addressed most of the previous comments in the following aspects:

1. Conceptual advantage of sub-SAGE compared to linear approach of feature selection

2. The advantage of sub-SAGE uncertainty calculating compared to SHAP.

3. Discussion of the advantage and limitations of sub-SAGE approximation compared to SAGE.

Here are some additional comments:

1. In lines 472-475, the authors discussed that the threshold and multiple testing procedure is needed to control the false positive rate. The authors should further expand this part since how to set the threshold and demonstrate calibration is not directly clear from the estimated uncertainty computed using the sub-SAGE approach, but a crucial step in GWAS.

2. In table 3, it would be good to run linear models on the selected feature, compute their feature coefficient and the corresponding p-values and discuss how much benefit we gain from computing feature importance in a nonlinear way using sub-SAGE.

3. In line 398, the SNP rs2820312 exclusively detected by sub-SAGE need further analysis. For example, have any other SNPs in the LD block of SNP rs2820312 been previously detected to be associated with obesity ?

4. In lines 450-454, the authors described several sources of the uncertainty of the Sub-SAGE uncertainty estimate. However, another important source of uncertainty, the model uncertainty, needs to be mentioned. I suggest the authors add it to the text.

Overall, the proposed algorithm is novel and could potentially benefit the understanding of large-genetic dataset. However, the advantage of sub-SAGE on actual genetic applications still has room for improvement.

**Have the authors made all data and (if applicable) computational code underlying the findings in their manuscript fully available?**

Reviewer #1: Yes

Reviewer #3: Yes

PLOS authors have the option to publish the peer review history of their article (what does this mean?). If published, this will include your full peer review and any attached files.

Reviewer #1: No

Reviewer #3: No

Figure Files:

Data Requirements:

Reproducibility:

References:

---

## [Decision Letter · Decision Letter 2]

20 Feb 2023

Dear Mr. Johnsen,

We are pleased to inform you that your manuscript 'Inferring feature importance with uncertainties with application to large genotype data' has been provisionally accepted for publication in PLOS Computational Biology.

Best regards,

Shaun Mahony

Academic Editor

PLOS Computational Biology

Ilya Ioshikhes

Section Editor

PLOS Computational Biology

Reviewer's Responses to Questions

**Comments to the Authors:**

Reviewer #3: The authors have addressed major concerns from the previous revision. Overall, this work presents an innovative algorithm that will motivate additional work in the community.

**Have the authors made all data and (if applicable) computational code underlying the findings in their manuscript fully available?**

Reviewer #3: Yes

PLOS authors have the option to publish the peer review history of their article (what does this mean?). If published, this will include your full peer review and any attached files.

Reviewer #3: No

---

## [Editor Report · Acceptance letter]

8 Mar 2023

PCOMPBIOL-D-22-01012R2 

Inferring feature importance with uncertainties with application to large genotype data

Dear Dr Johnsen,

I am pleased to inform you that your manuscript has been formally accepted for publication in PLOS Computational Biology. Your manuscript is now with our production department and you will be notified of the publication date in due course.

With kind regards,

Anita Estes
